# Effectiveness of Hypnoanalgesia in Paediatric Dermatological Surgery

**DOI:** 10.3390/children8121195

**Published:** 2021-12-17

**Authors:** Peláez Pérez Juana María, Sánchez Casado Marcelino, Quintana Díaz Manuel, Benhaiem Jean Marc, Escribá Alepuz Francisco Javier

**Affiliations:** 1Department of Anaesthesia, Division of Pediatria, Puerta del Mar University Hospital of Cádiz, University of Cádiz, 11009 Cádiz, Spain; 2Department of Intensive Care, Neurotrauma Critical Care Division, Toledo General University Hospital, University of Castilla-La Mancha, 41007 Toledo, Spain; mmsc16@gmail.com; 3Department of Intensive Care, Polyvalent Critical Care Division, Autónoma University of Madrid, 28046 Madrid, Spain; mquintanadiaz@gmail.com; 4Department of Anaesthesia, Medical Hypnosis Division, Faculty of Medicine of the Pitié-Salpêtrière, Sorbonne University, 75013 Paris, France; jmbenhaiem@wanadoo.fr; 5Department of Anaesthesia, Division of Pediatria, University and Polytechnic Hospital La Fe, University of Valencia, 46026 Valencia, Spain; escriba1982@gmail.com

**Keywords:** hypnosis, hypnoanalgesia, anxiety, pain, paediatric surgery, dermatology

## Abstract

Background and objective. Stress in surgical settings has subtle psychological and physiological repercussions in children. The objective is to evaluate whether hypnosedation is effective in reducing the doses of sedation and analgesia required during the periprocedural period in children undergoing dermatological surgery, without negatively affecting pain and satisfaction. Patients and methods: A prospective, longitudinal, observational study where paediatric patients (aged 5–16 years) scheduled for dermatological surgery were analysed according to whether they received hypnosis or distraction during surgery (both common procedures at the centre). As outcome measurements we used sedation doses (propofol) during surgery and the need for analgesia; pain assessment post-surgery and at 24 h using a visual analogue scale (VAS) or revised face pain scale (FPS-r) (both 0–10) depending on age, as well as patient and guardian satisfaction (on a scale of 0–10). Results: Of the 68 patients eligible during the follow-up period, 65 were included. Of these, 33 were treated with hypnosis and 32 with distraction. Children who underwent hypnosis required less total propofol (45.5 ± 11.8 mg vs. 69.3 ± 16.8 mg; *p* < 0.001) and metamizole in the immediate postoperative period (34.4% vs. 65.6%; *p* = 0.018). After 24 h, they required less ibuprofen (9.1% vs. 28.1%; *p* = 0.048) and paracetamol (48.5% vs. 75.0%; *p* = 0.028). Mean pain according to VAS or FPS-r at 24 h was 3.1 with hypnosis vs. 4.3 with distraction (*p* < 0.001). Overall satisfaction was higher in the hypnosis group (8.7 ± 0.1 vs. 8.1 ± 0.2; *p* = 0.009). Conclusions: Hypnoanalgesia in children undergoing dermatological outpatient surgery could not only reduce sedation and analgesia requirements, but also improve child and guardian(s) satisfaction.

## 1. Introduction

The diagnostic and therapeutic procedures involved in a child’s hospitalisation create fear and anxiety, and may lead to adverse changes in the child’s behaviour, which sometimes continue in the postoperative period [1,2]. Therefore, it is desirable to perform some kind of intervention to alleviate this fear and anxiety. It is in the patient’s best interest to alleviate or modulate the pain involved in surgery.

The use of pharmacological interventions, such as benzodiazepines or midazolam, seeks to reduce preoperative anxiety in children. However, this may be associated with undesirable effects, such as paradoxical reactions, prolonged sedation and adverse changes in behaviour [3,4,5].

Some non-pharmacological techniques encourage the patient’s attention to be directed away from anxiogenic stimuli and towards a ‘safe place’ [6]. These techniques include hypnosis, which is useful in focussing the patient’s attention and activating their own resources to lead them to a safe place, away from the anxiogenic stimulus. This enables them to actively collaborate in their own care and promotes the patient’s autonomy. Other settings may also be used to distract children’s attention during medical-surgical care [7].

Most high-quality studies on pain and hypnosis have been conducted in adults. Studies on pain and hypnosis, such as Thompson et al. [8], demonstrate the efficacy, safety and benefits of hypnosis with regards to pain as a non-pharmacological, integrative therapy in the adult population. The mechanisms of action proposed for non-pharmacological, integrative therapies in adults may differ in the paediatric population due to, among other reasons, the potential developmental effects, which means specific studies are needed in this population. Leading authors in the study of paediatric hypnosis, such as Olness [9] and Sugarman [10] show a greater response to hypnosis in children due to their creative thinking, especially between the ages of 7 and 14 years. A good therapeutic relationship is necessary, as is adapting hypnotic techniques to the child’s preferences, age and level of cognitive development.

Furthermore, hypnosis during anaesthesia has been shown to help reduce analgesic and sedative doses during and post-surgery, thereby facilitating recovery [11]. This is considered an advantage, as it helps prevent over-medication in children. Given the expected efficacy in children and adolescents due to their responsivity to hypnosis techniques, we can expect hypnoanalgesia to be effective in this population for minimally invasive interventions (catheter cannulation, probes and other interventions), or as an adjuvant treatment in patient sedation under local or locoregional anaesthetic [12]. In this study, hypnosis was proposed as an adjuvant treatment to sedation for dermatological surgery. The aim was to assess whether using this hypnoanalgesic technique in paediatric dermatological surgery reduces the need for periprocedural sedation and analgesia without negatively affecting either immediate and 24-h postoperative pain, or the child’s and/or their guardian’s satisfaction with the procedure.

## 2. Materials and Methods

### 2.1. Design

A prospective, longitudinal, observational study on the comparative effectiveness of interventions, in this case sedation, was conducted. The observation period covered the surgical procedure, immediate postoperative period and 24 h following surgery. The study was approved by the Clinical Research Ethics Committee at the Complejo Hospitalario de Toledo [Toledo Hospital Complex] and was carried out in accordance with the WHO code of ethics (Declaration of Helsinki) on human experimentation.

This study followed STROBE recommendations, as shown in Figure 1.

### 2.2. Participants

Children scheduled for outpatient dermatological surgery at Centro Nacional de Parapléjicos [National Paraplegic Centre] in order to perform a diagnostic or therapeutic biopsy, exeresis of benign and malignant lesions and the infusion of the botulinum toxin were selected based on the following inclusion criteria:

(1) having been classified as anaesthetic risk class I or II according to the American Society of Anaesthesiologists (ASA); (2) being in a height and weight percentile between P3 and P97; (3) having no known drug allergies; (4) fasting for 6 h (solids) and 2 h (liquids); and (5) having Spanish as their mother tongue. Those with: a diagnosed intellectual disability, attention deficit disorder, behavioural disorders, previous treatment with hypnosis, a history of neurological pathology or psychomotor delays, previous painful pathology and obstructive sleep apnoea (OSA) were excluded from the study.

The exclusion of patients was carried out bearing in mind that children suffering from an attention deficit disorder are incapable of selecting and concentrating on the relevant stimuli, those with prior pain or neurological disorders could disrupt the use of low doses of sedatives, those who have previously received hypnosis could have a preconceived idea; the obese could disrupt the distribution of propofol and moderate-severe asthmatics might need additional medication and present respiratory complications.

Recruitment was undertaken by the main researcher based on the surgery waiting list over a three-year period (November 2016 to November 2019). All patients who met the inclusion criteria received an informed consent form together with an information sheet on both the anaesthetic procedure and hypnosis as an adjuvant treatment technique.

### 2.3. Procedures and Interventions Compared

Figure 2 shows the global study procedure. Our centre uses both conventional sedation and clinical hypnosis for outpatient surgical procedures. The distribution of these techniques depended on the day of intervention, i.e., hypnosis or distraction technique day.

I Preoperative: recruitment of patients: II Intraoperative: depending on the day with hypnosis (intervention), the day with no hypnosis (attention distraction techniques), application in both arms of induction and maintenance anaesthesia by means of intravenous and/or inhalation agents or local anaesthetic and elimination; awakening (no hypnosis)/return to normal alertness (hypnosis) III. PACU control of vital signs and pain. IV Postoperative monitoring 24 h: pain control.

On the day of the intervention, the children were in the pre-operative room where we interacted with them to learn about their sensory capabilities, reduce anxiety and fear, and initiate the hypnosis technique with focus of attention consistent with the child’s preferences and appropriate to the neurocognitive level or focus of attention on images or music with the digital tablet. At that time, we gave the children fruit-scented markers to colour the anaesthetic face mask inside and out, in addition to applying EMLA^®^ anaesthetic cream on the back of both hands at least 60 min before peripheral venous cannulation.

#### 2.3.1. Compared Interventions

##### Hypnosis

In the hypnosis group, we used suggestion through metaphor that relies on the child’s imaginary thinking to change their perception. This is an excellent moment to use the expression “as if” within the imaginary world of the metaphor. In order to explain the therapeutic act to a seven-year-old child, we could say, for example, that we are going to use a “magic or special mask”, through which the scent of mint, apple or strawberry will enter the airway as if they were sweets that make you laugh while you travel to the moon.

Besides facilitating the emergence of hypnosis in the patient, this process enables the child to participate in the surgery. Following standard sedation, therapeutic suggestions are maintained throughout the surgical procedure and the posthypnotic period before returning the patient to the alert state of anaesthesia.

##### Distraction Techniques

The distraction technique used was a digital tablet, on which a cartoon or a music video was passively played as chosen by the child. As with hypnosis, it was maintained throughout the perioperative process, until post-anaesthesia awakening.

#### 2.3.2. Anaesthetic Procedure

##### Pre-Anaesthetic Consultation

In the pre-anaesthetic consultation, a therapeutic relationship was established with all children, regardless of the group. This was done by letting them choose their favourite experience of therapeutic suggestion, according to age, preference, and level of cognitive maturation. Furthermore, behavioural interventions were carried out with patients and guardians to reduce the anxiety and fear associated with the procedure, and eliminate any negative connotations associated with medical hypnosis. We explained to the child and guardians that the objective of these interventions was to reactivate the patient’s resources as a method of innate adaptation, creating a “safe place” in the surgical environment. In the case of hypnosis, we used the child’s visual, aural, kinaesthetic, olfactory and gustatory sensorial realities to create this “safe place”, which also helped them to concentrate on their interior world, their imaginary world which becomes their virtual reality. We must observe what the child’s main, predominant sensorial channel is, thus giving us a natural vision of how they perceive reality. The clinician who carries out the hypnosis will accompany the patient using obvious and trivial phrases (truisms), talking in a calm tone and rhythm of voice. In this way the patient shares their same reality and feels orientated in the “here and now”, in order to achieve their medical care objective. The patient’s autonomy is respected throughout the whole process. In essence, hypnosis is a type of therapeutic communication which improves adherence to medical care. When using the distraction technique by means of a digital tablet, it will be the child who chooses their distraction, which may be visual and/or aural and, once the tablet has been handed over, the procedure will be passive.

##### Intraoperatorion

(1)Anaesthetic Induction

Once the patient was in the operating theatre, they were monitored as per the Spanish Anaesthesia and Reanimation Society (SEDAR) for this procedure and ASA state, regardless of the technique used to focus attention, i.e., hypnosis or no hypnosis (distraction technique).

The child was allowed to choose between inhalational induction of a nitrous oxide and oxygen mixture (60%/40%) and intravenous induction. All but three children chose inhalational induction. A Mapleson^®^ paediatric external circuit was used for inhalation of this mixture connected to the MaquetFlow-i C20 anaesthesia machine and a scented face mask (except for the three children who rejected inhalational induction), which had to be maintained throughout a 3-min induction period prior to any intervention. This mask was connected to a 1 L reservoir bag with a pressure limiting valve (APL) enabling extraction of exhaled anaesthetic gases via the scavenging system. Fresh gas flow is 8 to 10 L as per the patient’s weight i.e., equal to 2–3 times their volume/minute, essential to prevent re-inhalation in Mapleson^®^ circuits. This system maintains spontaneous breathing throughout the procedure. Had the child not wanted inhalational anaesthetic induction, we would have used the local anaesthesia provided by the EMLA cream.

During inhalational or intravenous anaesthetic induction, only one person should interact with the child verbally, rather than the entire team, to prevent attention dispersion in both the distraction and hypnosis technique groups.

Once the peripheral venous cannula had been inserted and fluid-therapy commenced with Ringer’s lactate, standard intravenous induction was started with propofol at 1.5 mg/kg. Next, the surgeon injected 2% lidocaine local anaesthetic subcutaneously, followed by surgical incision and subsequent extraction of the skin lesion, haemostasis and suture.

(2)Maintenance

Inhalation of the nitrous oxygen/oxygen mixture was maintained throughout the intervention, and the patient was told to just breathe normally.

An additional 1.5 mg/kg propofol was administered based on vital signs during surgery which might lead to activation of the autonomous nervous system in response to a sympathetic nociceptive stimulation as a primary indicator of a second dose of propofol being required; moreover, if pain control was insufficient, short-acting opioids (Alfentanil 10–20 mcg/kg) were administered until stabilisation of vital parameters.

(3)Elimination

The nitrous oxide concentration is reduced by 25% near the end of the procedure to prevent nausea and vomiting. The patient wakes up in the operating theatre after surgery and gradually returns to a state of alertness when they are moved to the PACU (post-anaesthesia care unit).

The mean duration of the surgical interventions was 30–40 min.

##### Immediate Post-Operation

The PACU not only controls the patient’s vital signs but also assesses pain using adapted scales (see variables and measurements below), and analgesia was administered where necessary. Paracetamol, magnesium metamizole or intravenous tramadol were used.

##### Telephone Check 24 h Post-Operation

The nurse monitoring the amount of drugs administered in both groups as per adapted pain scale assessment performed the 24 h post-operation check. Paracetamol and ibuprofen were administered orally.

### 2.4. Evaluation and Outcome Measurements

The primary endpoint of the study was the total propofol dose (mg) and additional opioid requirement during surgery, measured in mg/kg body weight, as recorded intraoperatively.

Furthermore, the impact on pain and analgesic needs were compiled, both in the immediate postoperative period and 24 h post-surgery. These were measured using pain scales adapted to age and cognitive maturation, in this case the visual analogue scale (VAS) from age 10 and revised face pain scale (FPS-r) from five to 9 years (both with a scale of 0–10), as well as the need for paracetamol, ibuprofen or other analgesic. The scales were given to the children by the head PARU nurse, who was also responsible for the 24-h post-surgery telephone follow-up, and was unaware of the group to which each patient belonged. Analgesic need was recorded in the patient’s history.

Prior to discharge, the child and their guardian(s) were given a satisfaction questionnaire on the overall care provided by the healthcare staff during the outpatient paediatric dermatological procedure, with six questions on a Likert scale (1–4) and an overall satisfaction scale (0–10).

### 2.5. Statistical Analysis

For the descriptive analysis, mean and standard deviation were used for quantitative variables, and frequency tables (percentage distribution) were used for qualitative variables. The outcome measurements for the intervention control groups were compared using parametric tests for variables with a normal distribution, chi-square or Fisher’s exact test for qualitative variables, depending on the number of cases, and Student’s t-test for quantitative variables. The Mann-Whitney U test was used for quantitative variables that do not follow a normal distribution.

The significance level was set at a value of *p* < 0.05 and analysis was carried out with Stata 12.1 (College Station, TX, USA).

## 3. Results

A total of 68 children were recruited between November 2016 and November 2019. 65 of these children met the selection criteria and were included in the study, while three patients were excluded from the total, two due to obesity and one due to asthma. All patients who were included completed the study (Figure 2).

Indications for surgery were: nevus (n = 35), local neoplasms (n = 6), and other lesions (n = 24). Ages ranged from five to 16 years, with 50% aged eight or younger. All weighed 22 kg or more. No significant differences in baseline patient characteristics were observed between the two study groups (Table 1).

Table 2 shows the results for the three evaluation periods: during surgery, the immediate postoperative period and 24 h after surgery.

During surgery, a significant reduction of the intraoperative propofol dose is shown both in absolute value (24 mg less on average) as well as per kg of body weight (0.8 mg/kg less) in the hypnosis group, meaning it achieved statistical significance (*p* < 0.001) in both cases. In addition to propofol, three patients (5%) required alfentanil on one occasion in the control group (3%) and two children in the hypnosis group (6%). The difference in proportions was not statistically significant.

During the stay in the PARU, which was statistically shorter for children in the hypnosis group, 45 children (69%) required some analgesic. The overall need was lower in the hypnosis group (26 children [81%] than in the distraction group versus 19 [58%] in the hypnosis group required some analgesic; *p* = 0.039). Regarding the drugs used, both paracetamol and metamizole magnesium were administered; only one child required tramadol (in the distraction group). Nevertheless, immediate postoperative pain measured by VAS or FPS-r was similar in both groups.

However, during the stay in the PARU, nine children (28%) in the distraction group and two (6%) in the hypnosis group required antiemetics (*p* = 0.023).

41 children (63%) required analgesia in the 24 h following surgery. This requirement was 25% lower in the hypnosis group (*p* = 0.05) generally and statistically different in favour of the hypnosis group in terms of paracetamol or ibuprofen use. The total score on the surgical procedure satisfaction questionnaires did not differ between groups (12 in the distraction group compared to 11 in the hypnosis group; *p* = 0.409) nevertheless, the overall scale favoured the hypnosis group (*p* < 0.01)

## 4. Discussion

The study shows that medical hypnosis in minor outpatient surgery in the paediatric population allows the child to take part in their own medical care by respecting their autonomy and using their own resources, while neither increasing pain nor affecting satisfaction, thus forming part of a new experience or learning process. It enables the dose of drugs administered to be reduced intraoperatively, in the immediate postoperative period, and 24 h post-surgery.

The comparison group (distraction) was also shown to be effective for anxiety and pain with regard to invasive procedures in children [13]. As an intervention with a beneficial effect, evidence of the effectiveness of hypnosis is even more relevant, as it competes with a proven adjuvant treatment technique, and not only with no treatment [14].

The dimensions of the real and virtual worlds are present in hypnosis. The child focuses their attention on what interests them in such a way that the suggestion hinges on the imaginary thoughts of the child, and their imaginary world fluidly becomes their virtual reality [15]. The spontaneous trance is the focus and target of attention over a certain period of time. The provoked trance or hypnosis allows for the use of the positive spontaneous trance at certain times and places in the medical domain. The clinician who carries out the hypnosis is there to accompany and later guide the patient towards their normal state of alertness. The medical setting favours spontaneous trance in patients which means that it is necessary to “keep them under observation” as an induced trance is not always necessary. The hypnotic process makes it possible to recruit cerebral regions, such as the anterior cingulate cortex, which take part in pain management [16,17].

As opposed to hypnosis, distraction techniques, whether they be active or passive, focus the attention, in this case, on images or music on the tablet. In the case of medical care, this attention is not maintained in the same way over time and there is no permanent doctor-patient relationship. Unlike the accompaniment carried out during hypnosis, it does not use the patient’s resources and the cerebral areas implicated in the brain areas involved in neuromatrix pain.

Focusing solely on dermatology helped us to eliminate the methodological ‘noise’ of procedures, complications, the surgeon and technique. The surgical procedure corresponded to moderate pain, and preventive analgesia was used in both groups during the peri-procedure as were non-pharmacological techniques. Studies by Talour et al. [18] and Jones et al. [19] reveal predictive factors for pain in dermatological surgery, including the type of surgery, which can be associated with moderate pain (excision, biopsy, etc.) to very severe pain (corticosteroid injection, laser, etc.) pain; sex; age; previous experience of surgery or pain; preventive analgesia; use of pharmacological or non-pharmacological techniques; or surgical site. Assigning children to one group or the other could be considered quasi-randomised, since the day of the scheduled surgery was the sole variable; and this was proven by the lack of differences between the groups compared. This also enabled us to avoid having to perform multivariate analysis to control covariates.

Pain is more complex than simply a stimulus-response action. In the surgical process, we must consider the A delta fibres responsible for acute pain, and the C fibres, which are slow-conduction, non-myelinated nerve fibres. These fibres initially transmit pain, while cognitive and emotional aspects are involved in producing the painful sensation. The painful experience is initiated and maintained in specific brain structures, and its final outcome is determined by memory, cognitive state and emotional situation, hence the importance of acting on this sensory and emotional dimension. Initial studies by Faymonville et al. and more recent controlled studies in invasive procedures in adults [8] have demonstrated the efficacy of hypnoanalgesia.

Hypnosis is developed as a process that takes place in different stages, and begins with inviting the patient to a new experience, followed by the disassociation of consciousness where a change occurs in the patient’s perception, and ending with a return to their usual state of alertness.

The hypnotic state is a natural one (through fascination, such as by watching a sunset, for example) which a clinician using hypnosis can facilitate via suggestion. It draws on the patient’s resources and specific needs to encourage a change in perception. Hypnosis involves a change in the child’s perception until they return to their normal alert state. The hypnotic trance, which can be achieved immediately or within minutes, depends on the suggestibility of the individual, the clinician’s hypnosis skills, and the context [20]. In recent years, it has been postulated that hypnotic suggestions reduce pain by activating endogenous pain inhibitory systems which descend to the spinal cord, thereby preventing nociceptive information from being transmitted to the brain [21]. Naloxone does not reverse hypnotic analgesia, and as such, the mechanisms are not dependent on endogenous opioids [22].

However, comprehensive therapies do not necessarily work the same way in adults and children, likely due both to developmental effects as well as pharmacokinetic and pharmacodynamic differences. Based on these differences, we prioritised the use of short-acting drugs with a dissociative [23] and analgesic effect (like nitrous oxide) [24], a dissociative effect (like propofol), and an analgesic effect (such as alfentanil and lidocaine) in our study. In addition to lower analgesia requirements, there was also a decrease in nausea and vomiting in the immediate postoperative period in favour of the hypnosis group. This is a known beneficial effect of hypnosis in chemotherapy studies [25]. Nevertheless, the procedure is considered low risk as it is a short stay surgery, so nitrous oxide use was reduced by 25% before the end of the intervention; moreover, dermatological surgery is not considered emetic. Overall, our study supports the concept of “pain-free dermatology paediatrics” [26,27] with high media and paediatric impact, complementing the multidisciplinary care model and implementing the integrative medicine model [28,29,30].

The limitations of this study are those inherent to the hypnosis process; the patient’s collaboration is required, given that they are autonomous. We would have preferred to have conducted a clinical trial; however, as already mentioned, group assignation was quasi-randomised. Another study limitation was the fact that only one anaesthetist was in charge of both hypnosis and medication, but the hospital would not allow two anaesthetists to be involved. On the other hand, the sedation and analgesic protocol is very strict and the whole team was careful to avoid any deviation.

This study and the conclusions drawn from it would benefit from replication, which we hope others will undertake.

## 5. Conclusions

The use of hypnoanalgesia in paediatric outpatient surgical settings improves perioperative pain perception, decreases drug consumption, decreases recovery time and increases patient responsiveness, resulting in less physiological stress and greater patient satisfaction.

## Figures and Tables

**Figure 1 children-08-01195-f001:**
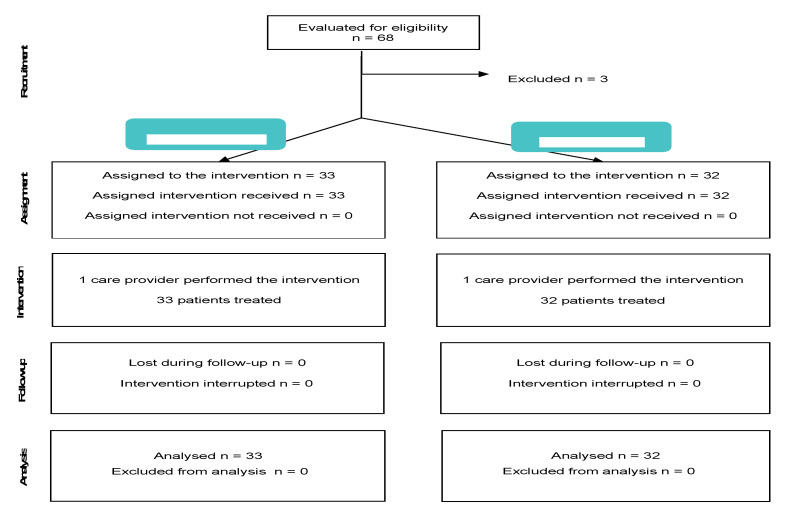
Study flowchart (STROBE).

**Figure 2 children-08-01195-f002:**
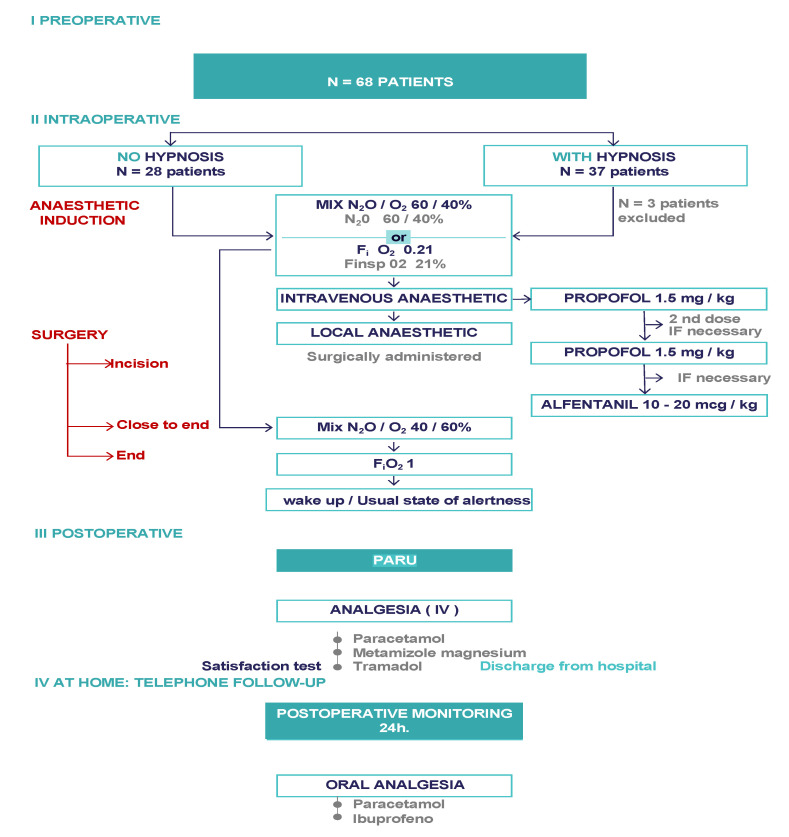
Study procedure and interventions compared: Hypnosis and attention distraction.

**Table 1 children-08-01195-t001:** Baseline characteristics of 65 children undergoing hypnosedation or distraction during major dermatological outpatient surgery.

	Control (n = 32)	Hypnosis (n = 33)	*p* Value
Age, m (SD)	8 (2)	8 (3)	0.905
Weight, m (SD)	29 (8)	28 (7)	0.571
ASA, n (%)			
I	28 (88)	28 (85)	0.520
II	4 (13)	5 (15)	
Surgical technique			
Exeresis	29 (91)	29 (88)	0.545
Biopsy	3 (9)	4 (12)	
Associated pathology, n (%)	4 * (13)	3 * (9)	0.708

Quantitative variables expressed as mean and standard deviation. Qualitative variables expressed as frequency (%). * Obesity and asthma.

**Table 2 children-08-01195-t002:** Outcome measurements and comparison between groups.

	Distraction (n = 32)	Hypnosis (n = 33)	*p* Value
Intraoperative sedation			
Propofol (mg), m (SD)	69.3 (16.8)	45.5 (11.8)	<0.001
Propofol (mg/kg), m (SD)	2.4 (0.6)	1.7 (0.4)	<0.001
Alfentanil, n (%)	1 (3.1)	2 (6.1)	0.512
Immediate postoperative period			
Paracetamol, n (%)	26 (81.3)	19 (57.6)	0.039
Metamizole, n (%)	21 (65.6)	12 (36.4)	0.018
Tramadol, n (%)	1 (3.1)	0 (0)	0.492
VAS or FPS-r (0–10), m (SD)	4.9 (1.5)	4.6 (1.4)	0.358
PARU stay (minutes), m (SD)	83.1 (21.9)	69.4 (16.0)	0.005
After 24 h			
Paracetamol, n (%)	24 (75.0)	16 (48.5%)	0.028
Ibuprofen, n (%)	9 (28.1%)	3 (9.1%)	0.048
VAS or FPS-r (0–10), m (SD)	4.40 (0.91)	3.25 (0.78)	0.001
Degree of satisfaction (0–10)	8.1 (1.2)	8.7 (0.7)	0.009

Abbreviations: m, mean; SD, standard deviation; VAS, visual analogue scale; FPS-r (revised Face Pain Scale); PARU, post-anaesthesia resuscitation unit.

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
