# Peer review of "Effectiveness of Hypnoanalgesia in Paediatric Dermatological Surgery"

_children, 2021, doi:10.3390/children8121195_

Round 1

Reviewer 1 Report

p 2 Lines 43 and 44:  The 'sentence' beginning "Therefore it is desirable..", and ending "involved in surgery" is not a proper sentence.  I suggest dividing to 2 sentences:  Starting with "Therefore..." ending with "...fear and anxiety." A second sentence would then be: " Similarly (delete "likewise"), it is in the patient's best interest to alleviate or modulate the pain involved in surgery."

p 2 Lines 61-64 While the statements here are true and accurate, the references could be better and more up-to-date.  Textbooks on Child Hypnosis and references therein could add substance and support to these affirmations.  I'd suggest the texts in general, the chapters about surgery and pain, and the extensive references that are easily accessible in two texts:  Therapeutic Hypnosis with Children and Adolescents 2nd edition - Editors: L.I. Sugarman MD and W.C. Wester II EdD 2014 Crown House Publishing and Hypnosis and Hypnotherapy with Children - 4th edition - Authors: D.P. Kohen, M.D. and Karen Olness, M.D. Routledge 2011

While reference #19 cited on line 64 is a fine paper about Ethical considerations, I'm not sure that it is a relevant reference in support of the sentence where it is mentioned.

p. 2 Line 68  "...due to their sensitivity to hypnosis..." does not seem to be the right word choice for what the authors seem to wish to convey.  I believe you (authors) will agree that you meant to say "due to their responsively to hypnosis".

p. 2 Line 70  I believe the use of "etc." is to be discouraged as it is likely meaningless and adds nothing to the average reader's understanding of what is being said.  Perhaps you would prefer "et al" or "and other interventions" to be more precise.  Here and anywhere else "etc." doesn't feel or seem appropriate to me.

p.3 Lines 97-101.  It's of course appropriate and wise to include those who were excluded from the study.  However I think it would strengthen the readers' understanding of the study to know the REASONS for exclusion of those categories / diagnoses, just as you carefully articulated the INCLUSION criteria.

p. 5 Lines 117-120 I would suggest re-writing these sentences as follows: 
"Besides facilitating the emergence of hypnosis in the patient, this process (rather than the word "it") enables the child to participate in the surgery. Following standard sedation, the therapeutic suggestions are maintained throughout the surgical procedure and post-hypnotic period prior to alerting the patient from anesthesia."

p. 5 Lines 128-9 - "behavioural intervention were carried out" would benefit from expansion and particularly for readers (surgeons, anesthesia professionals) who may be less familiar with behavioural interventions than others.  Brief examples of 2 or 3 'typical' or 'usual' behavioural interventions employed would be helpful.  In particular, a description of these should include some explanation as to how they would, as noted on Line 129, "eliminate any negative connotations associated with medical hypnosis."  In order to maximize the value of this important statement, use of examples and clarifying what negative connotations are of concern and HOW the behavioural interventions would obviate these concerns would be very helpful to the average reader.

p. 6 - Line 156 (*and 155 from preceding page) "the patient was instructed to breathe normally".  This may be so and maybe what has been 'usually' stated.  HOWEVER, the experienced clinician utilizing hypnosis with children would understand this as a hypnotic suggestion fraught with problems, though it is clearly not meant as a hypnotic suggestion! Why? Because "breathing normally" is what we do all the time WITHOUT thinking about it and an instruction to do so raises it to conscious level of thought and can be confusing...The very comment "just breathe normally" commonly evoke a CHANGE in breathing for this reason.

p. 6 Lines 158-180 are confusing to read because the authors switch from present tense to past tense to present.  Some of this seems to be because of an intent to teach HOW it should be done whereas this section is characterized as a description of the methodologies employed.

Line 158-159  "It should be in line with the child's preferences..." seems like it would be part of an instruction manual for clinicians but HERE I believe it should read "It was consistent with the child's preferences...."

Line 163-167.  Similarly, I'd suggest re-writing this segment as:  "The administration of ...propofol. at a dose of.." WAS based on vital signs during surgery.  This may lead to an increase in heart rate......propofol is needed." "If pain control WAS insufficient, short-acting opioids...were/was administered until ....are stabilized."

All of that said, it is not at all clear how relevant  lines 163-169 are to the study.  That is, please ask yourselves "IS this information relevant to what the reader needs to know and understand regarding the nature of the research and the outcomes achieved?"  If not, then they could be deleted.

Line 175  "The average duration....is 30 to 40 minutes" should be changed to "was", as it is a description of what took place in the study = past tense.

A generic kind of question that those skilled in clinical hypnosis with children may ask is something like "HOW was/is 'distraction' different from being a form of hypnosis SINCE both hypnosis as you are describing it and distraction both involve selective inattention or dis-attention with the goal of decreasing discomfort.   In this way one might consider a comparable study that included NO intervention as a control.

p.6  Evaluation and Outcome measures Lines 181-196 - and also p 7 Lines 197-224 -  Very nicely done and explained.  There is an unfamiliar word in the table= "Exeresis"  - "Google" defines it as surgical removal of an organ. As a pediatrician for over 50 years I never heard this word before, so I imagine others may have the same question?!

p. 9 Line 275 "...the child's co-peration" is a misspelling and should be, I believe, "cooperation".  However, it's important to explain and discuss in what way - HOW? - does the study  show that medical hypnosis facilitates cooperation? Was there a 'measure' of cooperation?

Line 284 - as noted earlier, suggest eliminate "etc."

Line 288-89 - I think "...to very severe (corticosteroid...)pain" should have the word pain after severe and before the parentheses:  thus in revising the sentence starting on 288: "....which can be associated with moderate pain (excision, biopsy, and others) to very severe pain (corticosteroid...) ...or surgical site."

Line 292 - at the end of the sentence "...surgery was was the sole..." obviously one of the "was" should be deleted!

Line 307  I suggest change " which a hypnotherapist can induce via suggestion." to "which a clinician using hypnosis can facilitate via suggestion" .  

p. 10 Lines 308-310  In keeping with the current understanding and nomenclature of Hypnosis with Children (See recent publications in CHILDREN on Hypnosis - State of the Art ) I suggest that the sentence beginning on line 308 as "Hypnosis develops as a process..." be changed to read: "Hypnosis develops as a process occurring in stages from beginning (initiation or invitation not "induction"), and dissociation of consciousness eating to the changing in perception to returning to usual state of alertness."  Note please that "confusion" is neither necessary, nor usually appropriate nor in any way typically or usually utilized with children as part of an intentional therapeutic hypnosis process.  Further it is not appropriate to refer to the absence of or ending of hypnosis as "returning to the waking state, BECAUSE hypnosis is NOT sleep, so it doesn't "end" by returning to being in a "waking" state.  Best is to refer to the child's usual state of alertness or awareness.

Line 312 - suggest change "....the hypnotherapist's skills...."  to "the clinician hypnosis skills and context."  It should be sgenerally understood that physicians who use hypnosis to help their patient are NOT "hypnotherapists" any more than physicians who prescribe penicillin to help their patients are "penicillinologists" !!

Line 323-324  "This is a known effect of hypnosis in chemotherapy studies."  Wouldn't it be more accurate to frame this as a known BENEFIT of hypnosis than as a known "effect" ?

p. 12 - Lines 396 and 403 - The references in #12 and #16 to Kuttner L., are identical and should be only one or the other

p. 12 - Lines 398 and 406 - The references in #14 and #18 Faymonville are identical...should only be mentioned once, not twice.

p. 13 - Line 436 - This reference is different from all others in being in all capital letters and should be fixed.

Author Response

Response to Reviewer 1

Thank you very much for your comments.

Point 1:

p 2 Lines 43 and 44: The 'sentence' beginning "Therefore it is desirable..", and ending "involved in surgery" is not a proper sentence. I suggest dividing to 2 sentences: Starting with "Therefore..." ending with "...fear and anxiety." A second sentence would then be: " Similarly (delete "likewise"), it is in the patient's best interest to alleviate or

modulate the pain involved in surgery."

Response 1: I accept change

Point 2:

p 2 Lines 61-64 While the statements here are true and accurate, the references could be better and more up-to-date. Textbooks on Child Hypnosis and references therein could add substance and support to these affirmations. I'd suggest the texts in general, the chapters about surgery and pain, and the extensive references that are easily accessible in two texts: Therapeutic Hypnosis with Children and Adolescents 2nd edition - Editors: L.I. Sugarman MD and W.C. Wester II EdD 2014 Crown House Publishing and Hypnosis and Hypnotherapy with Children - 4th edition - Authors: D.P. Kohen, M.D. and Karen Olness, M.D. Routledge 2011

While reference #19 cited on line 64 is a fine paper about Ethical considerations, I'm not sure that it is a relevant reference in support of the sentence where it is mentioned.
Response 2: I accept change

Point 3 :

p. 2 Line 68 "...due to their sensitivity to hypnosis..." does not seem to be the right word choice for what the authors seem to wish to convey. I believe you (authors) will agree that you meant to say "due to their responsively to hypnosis".

Response 3: I accept change

Point 4:

p. 2 Line 70 I believe the use of "etc." is to be discouraged as it is likely meaningless and adds nothing to the average reader's understanding of what is being said. Perhaps you would prefer "et al" or "and other interventions" to be more precise. Here and anywhere else "etc." doesn't feel or seem appropriate to me

Response 4: I accept change

Point 5:

p.3 Lines 97-101. It's of course appropriate and wise to include those who were excluded from the study. However I think it would strengthen the readers' understanding of the study to know the REASONS for exclusion of those categories / diagnoses, just as you carefully articulated the INCLUSION criteria

Response 5: I accept change

Point 6:

p. 5 Lines 117-120 would suggest re-writing these sentences as follows:
"Besides facilitating the emergence of hypnosis in the patient, this process (rather than the word "it") enables the child to participate in the surgery. Following standard sedation, the therapeutic suggestions are maintained throughout the surgical procedure and post-hypnotic period prior to alerting the patient from anesthesia."

Response 6 : I accept change . If I can't use the word return just like that, it is necessary to say return to the usual state of alertness. However for the distraction if I keep the post anaesthesia awakening, also in figure 2

Point 7:

p. 5 Lines 128-9 - "behavioural intervention were carried out" would

benefit from expansion and particularly for readers (surgeons, anesthesia professionals) who may be less familiar with behavioural interventions than others. Brief examples of 2 or 3 'typical' or 'usual' behavioural interventions employed would be helpful. In particular, a description of these should include some explanation as to how they would, as noted on Line 129, "eliminate any negative connotations associated with medical hypnosis." In order to maximize the value of this important statement, use of examples and clarifying what negative connotations are of concern and HOW the behavioural interventions would obviate these concerns would be very helpful to the average reader.

Response 7: I accept change .
Point 2.3.2 of procedure and comparison of interventions was very unclear, especially, as you say, for people who have no knowledge of hypnosis. This part has been rewritten. This explanation is given in the pre-anaesthesia consultation.

Point 8:

p. 6 - Line 156 (*and 155 from preceding page) LINE 270 "the patient was instructed to breathe normally". This may be so and maybe what has been 'usually' stated. HOWEVER, the experienced clinician utilizing hypnosis with children would understand this as a hypnotic suggestion fraught with problems, though it is clearly not meant as a hypnotic suggestion! Why? Because "breathing normally" is what we do all the time WITHOUT thinking about it and an instruction to do so raises it to conscious level of thought and can be confusing...The very comment "just breathe normally" commonly evoke a CHANGE in breathing for this reason.

Response 8: I accept change .

Point 9 : p. 6 Lines 158-180 are confusing to read because the authors switch from present tense to past tense to present. Some of

this seems to be because of an intent to teach HOW it should be done whereas this section is characterized as a description of the methodologies employed.

Response 9: This point 2.3 procedures and comparison of interventions has been structured in order to better explain HOW the procedure has been done.

Point 10:

Line 158-159 "It should be in line with the child's preferences..." seems like it would be part of an instruction manual for clinicians but HERE I believe it should read "It was consistent with the child's preferences... Response 10: I accept change. We have removed the sentence, when rewriting this section in preanaesthesia consultation

Point 11: Line 163-167. Similarly, I'd suggest re-writing this segment as: "The administration of ...propofol. at a dose of.." WAS based on vital signs during surgery. This may lead to an increase in heart rate......propofol is needed." "If pain control WAS insufficient, short- acting opioids...were/was administered until ....are stabilized."

All of that said, it is not at all clear how relevant lines 163-169 are to the study. That is, please ask yourselves "IS this information relevant to what the reader needs to know and understand regarding the nature of the research and the outcomes achieved?" If not, then they could be deleted.

Response 11:
I accept change to past time
I would not suppress it, the use of these doses of drugs seems relevant to me for a surgical gesture that associates several modalities of iv drugs, inhalations, cream solution together with the association of hypnosis or detraction of attention, I would ask myself HOW AND HOW MUCH?

Point 12 :

Line 175 "The average duration....is 30 to 40 minutes" should be changed to "was", as it is a description of what took place in the study = past tense

Response 12 : I accept change
Point 13
A generic kind of question that those skilled in clinical hypnosis with children may ask is something like "HOW was/is 'distraction' different from being a form of hypnosis SINCE both hypnosis as you are describing it and distraction both involve selective inattention or dis- attention with the goal of decreasing discomfort. In this way one might consider a comparable study that included NO intervention as a control.

Response 13.
Hypnosis is not the same as a distraction technique. In fact, I consider the control group as a distraction that has already been used in others.
In both there is a selective attention that is initiated and maintained and ends in a different way, I explain this in point 2. 3 and I also clarify it in the discussion. I have written that the use of the distraction technique is high-tech and passive; this means that we are not going to participate by interacting with the video or the music.

Point 14

p.6 Evaluation and Outcome measures Lines 181-196 - and also p 7 Lines 197-224 - Very nicely done and explained. There is an unfamiliar word in the table= "Exeresis" - "Google" defines it as surgical removal of an organ. As a pediatrician for over 50 years I

never heard this word before, so I imagine others may have the same question?!

Response 14
Exeresis surgical removal of a lesion in an organ, in this case, the skin. I do find it in Spanish but if you do not consider to be correct in English you can remove it , the surgeons I work with write it all the time and I have asked whether they know it as I do , but if you think it is better to use other words we can change it.

Point 15p. 9 Line 275 "...the child's co-peration" is a misspelling and should be, I believe, "cooperation". However, it's important to explain and discuss in what way - HOW? - does the study show that medical hypnosis facilitates cooperation? Was there a 'measure' of cooperation?

Response 15 : I accept change

Indeed, I don't think the word co-operation is the most appropriate. What I meant is that by using the child's resources, he/she adheres better to medical care, for him/her it is a new experience and a new learning from his/her imaginative thinking. I have re-written it in the discussion.

Point 16 Line 284 - as noted earlier, suggest eliminate "etc." Response 16: I accept change

Point 17 Line 288-89 - I think "...to very severe (corticosteroid...)pain" should have the word pain after severe and before the parentheses: thus in revising the sentence starting on 288: "....which can be associated with moderate pain (excision, biopsy, and others) to very

severe pain (corticosteroid...) ...or surgical site

Response 17: I accept change
Point 18 Line 292 - at the end of the sentence "...surgery was was the

sole..." obviously one of the "was" should be deleted!

Response 18 acepto cambio

Point 19 Line 307 I suggest change " which a hypnotherapist can induce via suggestion." to "which a clinician using hypnosis can facilitate via suggestion" .

Response 19 : I accept change

Point 20 p. 10 Lines 308-310 In keeping with the current understanding and nomenclature of Hypnosis with Children (See recent publications in CHILDREN on Hypnosis - State of the Art ) I suggest that the sentence beginning on line 308 as "Hypnosis develops as a process..." be changed to read: "Hypnosis develops as a process occurring in stages from beginning (initiation or invitation not "induction"), and dissociation of consciousness eating to the changing in perception to returning to usual state of alertness." Note please that "confusion" is neither necessary, nor usually appropriate nor in any way typically or usually utilized with children as part of an intentional therapeutic hypnosis process. Further it is not appropriate to refer to the absence of or ending of hypnosis as "returning to the waking state, BECAUSE hypnosis is NOT sleep, so it doesn't "end" by returning to being in a "waking" state. Best is to refer to the child's usual state of alertness or aware

Response 20
In order to explain the hypnotic process I think it was unfortunate to talk about confusion because it is also about children.
I have tried to explain it better in point 2.3 and in the discussion

Indeed, hypnosis is not sleep and wake up. In figure 2 we have written awakening/return, too little explanatory, you are right. Waking up for the distraction technique and return to the usual state of alertness for hypnosis, it is necessary to write it because it is important

Point 21

Line 312 - suggest change "....the hypnotherapist's skills...." to "the clinician hypnosis skills and context." It should be sgenerally understood that physicians who use hypnosis to help their patient are NOT "hypnotherapists" any more than physicians who prescribe penicillin to help their patients are "penicillinologists" !!

Response 21 : I accept change

Point 22

Line 323-324 "This is a known effect of hypnosis in chemotherapy studies." Wouldn't it be more accurate to frame this as a known BENEFIT of hypnosis than as a known "effect".

Response 22 : : I accept change
ITEMS 23 TO 25 ARE ALL LITERATURE REVIEWS THAT HAVE

BEEN ACCEPTED.

Point 23 p. 12 - Lines 396 and 403 - The references in #12 and #16 to Kuttner L., are identical and should be only one or the other
Point 24. p12 - Lines 398 and 406 - The references in #14 and #18 Faymonville are identical...should only be mentioned once, not twice. Point 25 p. 13 - Line 436 - This reference is different from all others in being in all capital letters and should be fixed.

Reviewer 2 Report

Excellent effort tackling a challenging research topic - hypnotic interventions in a clinical setting.  I have several comments/concerns, many relatively minor and some a bit more substantial:

Line 49: not all non-pharmacologic approaches are about attention/distraction (massage, acupuncture, etc).  I would add "Some" at the beginning of the sentence.

Line 51: Delete "diverting the patient's attention" as this signifies distraction.  Hypnosis is distinct from distraction techniques for exactly the reason you state next.

Lines 67-71.  I would argue that children are not more "sensitive" to hypnosis, but are more developmentally capable of utilizing it for themselves.

Line 92: Please provide examples of the types of dermatologic surgery included.  This is mentioned in the discussion but needs to be presented sooner.

Section "Comparative procedures and interventions: This section is confusing and needs significant re-writing. It is not clear to me exactly how your two arms are different.  Perhaps have two sub-sections - one which explains what happens on a "hypnosis day" and one which explains what happens on a "distraction" day. 

Later in this section you describe "a story was suggested."  This also needs clarification.  A story is not necessarily a "suggestion," which is a very specific, core component of hypnosis.  A suggestion can be direct or indirect and directly ties the child's goals for the hypnosis with the story/metaphor/state of relaxation evoked.  

Section "Evaluation and..."  I can see how the PARU nurse was blinded to the study groups, but it seems the anesthesiologist were not.  This is crucial to point out because it is a potential source of bias, especially if the anesthesiologists are aware of the study and the study team.  Since your primary endpoint is propofol and opioid use, which were controlled by the anesthesiologist, this is key.

Discussion section:  Why was a no-treatment control group not included?  There may be very good reasons for this, but they need to be mentioned.  The sentence "as an intervention with a positive effect..." is confusing and needs to be re-written.

Line 306-307: The hypnotic state is achieved by the process of inducing, and is then utilized by offering suggestions.  Suggestions do not induce hypnosis.  Please rewrite this sentence.

Line 308-309: Hypnotic induction and intensification sometimes utilizes confusion and/or dissociation, but not necessarily.  This should be re-worded.  It is certainly a process - one of inducing a focused state of concentration during which suggestions are offered which are directed at achieving mutually decided goals.  This may or may not involve cognitive distortion (confusion), dissociation, relaxation, etc.  Also, the word "waking" is misleading.  Hypnosis does not intentionally involve sleep (unless that's a stated goal).  I would suggest, perhaps, "re-alerting" anywhere you use "waking" in your paper.

Author Response

Response to Reviewer 2

Thank you very much for your comments

Point 1:

Line 49: not all non-pharmacologic approaches are about attention/distraction (massage, acupuncture, etc). I would add "Some" at the beginning of the sentence

Response 1: I accept change

Point 2:

Line 51: Delete "diverting the patient's attention" as this signifies distraction. Hypnosis is distinct from distraction techniques for exactly the reason you state next.

Response 2: I accept change
Point 3 : Lines 67-71. I would argue that children are not more "sensitive" to

hypnosis, but are more developmentally capable of utilizing it for themselves

Response 3: I accept change

Point 4:

Line 92: Please provide examples of the types of dermatologic surgery included. This is mentioned in the discussion but needs to be presented sooner

Response 4: I accept change

Point 5 : Section "Comparative procedures and interventions: This section is confusing and needs significant re-writing. It is not clear to me exactly how your two arms are different. Perhaps have two sub-sections - one which explains what happens on a "hypnosis day" and one which explains what happens on a "distraction" day.

Reponse 5: I accept change This point has been rewritten because we think that in the procedure (punto 2.3) it should be clarified in parts for a better understanding.

Point 6: Later in this section you describe "a story was suggested." This also needs clarification. A story is not necessarily a "suggestion," which is a very specific, core component of hypnosis. A suggestion can be direct or indirect and directly ties the child's goals for the hypnosis with the story/metaphor/state of relaxation evoked.

Response 6: I accept change. It has been rewritten 2.3.1.1 with an example using the metaphor

Point 7: Section "Evaluation and..." I can see how the PARU nurse was blinded to the study groups, but it seems the anesthesiologist were not. This is crucial to point out because it is a potential source of bias, especially if the anesthesiologists are aware of the study and the study team. Since your primary endpoint is propofol and opioid use, which were controlled by the anesthesiologist, this is key

Response 7: This is a limitation of the study but the hospital would not accept two anaesthetists being present in the operating theatre for this procedure, however, we proceeded with the maximum rigour within the protocol of analgesia and sedation, avoiding any deviation. I have written this at the end of the discussion

Point 8: Discussion section: Why was a no-treatment control group not included? There may be very good reasons for this, but they need to be mentioned. The sentence "as an intervention with a positive effect..." is confusing and needs to be re-written.

Response 8: Hypnosis is not the same as a distraction technique. In fact, I consider the control group as a distraction that has already been used in others.
In both there is a selective attention that is initiated and maintained and ends in a different way, I explain it in point 2. 3 and I also clarify it in the discussion. I have written that the use of the distraction technique is high- tech and passive; this means that we are not going to participate by interacting with the video or the music.

Point 9 : Line 306-307: The hypnotic state is achieved by the process of inducing, and is then utilized by offering suggestions. Suggestions do not induce hypnosis. Please rewrite this sentence.

Response 9: I accept change.

Rewritten in point 2.3.1.1 Hypnosis ....

Point 10: Line 308-309: Hypnotic induction and intensification sometimes utilizes confusion and/or dissociation, but not necessarily. This should be re- worded. It is certainly a process - one of inducing a focused state of concentration during which suggestions are offered which are directed at achieving mutually decided goals. This may or may not involve cognitive distortion (confusion), dissociation, relaxation, etc. Also, the word "waking" is misleading. Hypnosis does not intentionally involve sleep (unless that's a stated goal). I would suggest, perhaps, "re-alerting" anywhere you use "waking" in your paper

Response 10 : I accept change

With the desire to explain the hypnotic process I think it was unfortunate to talk about confusion because it is also about children.
I have tried to explain it better in point 2.3 and in the discussion
Indeed, hypnosis is not sleep and wake up. In figure 2 we have written awakening/return, too little explanatory, you are right.

Waking up for the distraction technique and return to the usual state of alertness for hypnosis, it is necessary to write it because it is important.

Round 2

Reviewer 1 Report

While nearly every comment was followed by "I agree" I did not see the agreement or changes in the revised text.  An example of this is that recommended expert references noted in the review are not cited or mentioned.  Moreover, apparently pointing out the duplicate listing of a specific reference (one of the Kuttner references is mentioned twice) was either ignored or perhaps inadvertently missed as this same reference is again mentioned twice in the references.  While the author(s) stated "I agree" after each comment, that agreement does not appear to have been translated into changes in the text.  

Author Response

Dear reviewer 1, I thank you for your comments and apologize that not everything was corrected.

One of the unclear parts we thought should be rewritten in the procedure section. This has caused me to change the line numbering and for this reason I am enclosing the manuscript responding point by point. Best regards

Juana M Peláez
